# Can Anhedonia Be Considered a Suicide Risk Factor? A Review of the Literature

**DOI:** 10.3390/medicina55080458

**Published:** 2019-08-09

**Authors:** Luca Bonanni, Flavia Gualtieri, David Lester, Giulia Falcone, Adele Nardella, Andrea Fiorillo, Maurizio Pompili

**Affiliations:** 1Psychiatry Residency Training Program, Faculty of Medicine and Psychology, Sapienza University of Rome, 00189 Rome, Italy; 2Psychology Program, Stockton University, Galloway, NJ 08205, USA; 3Department of Psychiatry, University of Campania Luigi Vanvitelli, 80138 Naples, Italy; 4Department of Neurosciences, Mental Health and Sensory Organs, Suicide Prevention Center, Sant’Andrea Hospital, Sapienza University of Rome, 00189 Rome, Italy

**Keywords:** anhedonia, PTSD, suicide, suicide risk, suicidal behavior

## Abstract

*Background and Objectives*: At present, data collected from the literature about suicide and anhedonia are controversial. Some studies have shown that low levels of anhedonia are associated with serious suicide attempts and death by suicide, while other studies have shown that high levels of anhedonia are associated with suicide. *Materials and Methods*: For this review, we searched PubMed, Medline, and ScienceDirect for clinical studies published from 1 January 1990 to 31 December 2018 with the following search terms used in the title or in the abstract: “anhedonia AND suicid*.” We obtained a total of 155 articles; 133 items were excluded using specific exclusion criteria, the remaining 22 articles included were divided into six groups based on the psychiatric diagnosis: mood disorders, schizophrenia spectrum disorders, post-traumatic stress disorder (PTSD), other diagnoses, attempted suicides, and others (healthy subjects). *Results*: The results of this review reveal inconsistencies. Some studies reported that high anhedonia scores were associated with suicidal behavior (regardless of the diagnosis), while other studies found that low anhedonia scores were associated with suicidal behavior, and a few studies reported no association. The most consistent association between anhedonia and suicidal behavior was found for affective disorders (7 of 7 studies reported a significant positive association) and for PTSD (3 of 3 studies reported a positive association). In the two studies of patients with schizophrenia, one found no association, and one found a negative association. For patients who attempted suicide (undiagnosed), one study found a positive association, one a positive association only for depressed attempters, and one a negative association. *Conclusions*: We found the most consistent positive association for patients with affective disorders and PTSD, indicating that the assessment of anhedonia may be useful in the evaluation of suicidal risk.

## 1. Introduction

Suicide is a severe public health problem with more than one million deaths reported each year worldwide [1] and with nearly 20 times more people attempting suicide each year [2]. Moreover, suicide ranks among the ten major causes of death worldwide and is a leading cause of death among youth and young adults in many countries [3,4,5,6]. Given the prevalence of suicidal ideation and behaviors and the associated medical expenses, it is important to know what factors lead to both suicidal thoughts and suicide attempts in order to identify reliably those transitioning from suicidal ideation to fatal suicidal behaviors [7]. Developing this ability will inform suicide risk assessment methods and early intervention tools.

The MATRICS consensus conference on negative symptoms, convened under the auspices of the National Institutes of Mental Health (NIMH) in 2005, suggested that there are five general categories of negative symptoms: avolition, anhedonia, asociality, affective blunting, and alogia. The most common negative symptoms are avolition and anhedonia, and one of the most persistent negative symptoms is anhedonia, that is not experiencing pleasure or having the ability to obtain pleasure from activities or interpersonal relationships. In DSM-5, anhedonia is defined as “Lack of enjoyment from, engagement in, or energy for life’s experiences; deficits in the capacity to feel pleasure and take an interest in things.” Anhedonia is present in both patients with schizophrenia and patients with depression. It is estimated that half of schizophrenia patients experience anhedonia, although estimates of its prevalence vary greatly. Anhedonia and social isolation predict schizophrenia in high-risk populations [8].

Lately, there has been a growing interest in anhedonia with the evidence suggesting that this symptom may be a unique predictor of psychopathology [9], including suicidality [10,11,12]. For example, a study of 40 patients with depression and 20 controls showed that individuals who had suicidal ideation and previous suicidal attempts were less likely to respond to rewarding stimuli on a task designed to assess the presence of anhedonia [12]. 

The results of the research, however, are inconsistent. Some research has shown that low levels of anhedonia are associated with serious suicide attempts and death by suicide [13,14], while other research has shown that high levels of anhedonia are associated with suicide [15,16,17,18]. A study conducted by Winer et al. [19] showed that anhedonia is associated with suicidal ideation but not with previous suicide attempts. Regarding the observed relationship between anhedonia and suicidal behavior, Nock and Kazdin [10] suggested that anhedonia is experienced as an intolerable state leading to suicidal behavior to escape a current stressor. Other research shows that those who think of suicide feel less pleasure and are focused on efforts to avoid psychological pain [12], and a reduction in psychological pain has been observed when individuals engage in suicidal behavior [20]. Some studies have suggested that anhedonia is a modifiable clinical factor associated with suicidal ideation, independently of depressive symptoms [21]. 

As mentioned above, several studies have shown that anhedonia is related to an increased risk of suicide, while others have claimed the opposite. This discrepancy can be understood using the following considerations [22]. First, anhedonia can be considered either as a trait or as a state. When anhedonia is a state, presenting and persisting as a symptom of a specific psychiatric disorder and, particularly, depression, it may be a risk factor for suicide. When it is considered as a trait, with long-term stability and not associated with depression, it may not act as a risk factor for suicide or it may be associated with a low risk of suicide [23]. Also significant is the difference between consummatory and anticipatory anhedonia [24], although this is rarely taken into consideration. A deficit in consummatory pleasure can characterize endogenomorphic depression, a subtype of depression associated with a higher risk of suicide [25]. Moreover, there is not just one but several types of suicide risk: suicidal ideation, suicide attempt (recent or not), and completed suicide. In addition, there is a great deal of difference in the methodology used in studies proposing to demonstrate a correlation between anhedonia and suicide risk which should be kept in mind. In fact, these studies do not have homogeneity in the types of sample chosen (differentiating for example by age, gender, etc.,) as well as a control of the confounding variables (such as outpatients or inpatients) and a control for depression level (that is, the effect of the anhedonia independently of depression) [22].

Although some medications, such as lithium [26,27] and clozapine [27,28], as well as psychotherapy, have shown effectiveness in managing suicidal behavior, the rates of suicidal ideation, suicide attempts, and completed suicides have not substantially decreased in recent years [29]. Hence, it will be useful to know in more detail the risk factors for suicide, such as anhedonia, in order to create alternative suicide prevention tactics, and the present review of the literature tries to understand the association between suicidal behaviors and anhedonia.

## 2. Materials and Methods

For this review, we searched PubMed, Medline, and ScienceDirect for clinical studies published from 1 January 1990 to 31 December 2018 with the following search terms used in the title or in the abstract: “anhedonia AND suicid*”. Initially, two reviewers screened the abstracts of studies identified for eligibility with 100% agreement, and then another reviewer assessed the included studies. All three reviewers discussed any disagreements identified and reached an agreement before proceeding. We obtained a total of 155 articles of which 133 articles were excluded using the following exclusion criteria: reviews or meta-analyses (21), case reports (2), letters/opinions (1), non-psychiatric studies (6) or unfocused (98), and studies conducted on adolescents (5). The 22 articles included in this review were divided into six groups based on the psychiatric diagnosis: mood disorders, schizophrenia spectrum disorders, post-traumatic stress disorder, other diagnoses (such as anxiety disorders), attempted suicide, and others (that is, without a psychiatric diagnosis). Details of the trials included (study design, sample size, criteria, objectives, evaluation tools, outcomes) were extracted and are reported in Table 1.

## 3. Results

### 3.1. Affective Disorders

In a retrospective longitudinal study, Ballard et al. [21] studied 100 treatment-resistant patients with major depressive disorder (*n =* 65) or bipolar disorder (*n =* 35) using data from several clinical trials of ketamine to evaluate whether anhedonia can be considered a clinical correlate of suicidal thoughts. At baseline, anhedonia measured both by the Snaith–Hamilton Pleasure Scale (SHAPS) scale and the anhedonia subscale of the Beck Depression Inventory (BDI) was associated with suicidal ideation. A reduction in suicidal thoughts 230 min after the administration of ketamine was associated with a reduction in anhedonia, independently of a reduction in depressive symptoms. The change in anhedonia accounted for an additional 13% of the variance for changes in suicidal thoughts beyond the effects of depressive symptoms. 

In a prospective longitudinal study of 187 adult patients with suicidal and/or non-suicidal self-injury history, Zielinsky and colleagues [30] found that anhedonia scores were higher in those with prior suicide attempts than in those with a history of non-suicidal self-injury. Depressive symptoms predicted suicidal behavior directly and also via mediation through anhedonia. 

Ballard and colleagues [31] carried out a prospective study in order to identify which clinical dimensions (suicidal ideation, loss of interest, anxiety, psychomotor agitation, high-risk behavior) increase before suicidal behavior. Data were collected from the Systematic Treatment Enhancement Program for Bipolar Disorder (STEP-BD) study. All the patients included were affected by bipolar disorders. Ballard et al. compared the participants who attempted suicide or completed suicide (*n =* 103) with patients without suicidal behavior (*n =* 427). The results indicated that patients with suicidal behavior had an elevation of all symptoms, including loss of interest. Moreover, suicidal ideation and loss of interest were highly intense in the months before the suicidal behavior so that they could be considered to be a risk factor for suicidal behavior in bipolar patients. 

In a cross-sectional study [12], Xie et al. explored the contribution of anhedonia and avoidance of pain as central factors in influencing the suicidal mind in a cohort of 40 depressed outpatients and 20 healthy control subjects. In the Affective Incentive Delay (AID) task, the High Suicidal Ideation (HIS) group had longer response times (RTs) under the reward condition than under the punishment condition, while the Low Suicidal Ideation (LSI) group and control groups had shorter RTs under the reward condition than under the neutral condition. The LSI group had also shorter RTs under the reward condition than under the punishment condition. Moreover, pain arousal and scores on the Beck Scale for Suicide Ideation (BSS) were negatively associated with differences in RTs between neutral and reward conditions, while pain avoidance and BSS scores were positively associated with differences in RTs between neutral and punishment conditions. Xie et al. concluded that the AID task was the best task for the detection of a hedonic approach to experience and of pain avoidance so that a reduced motivation to experience pleasure and an increased motivation to avoid pain could be considered reliable predictors of suicidal behavior. 

In order to explore determinants of suicidal ideation and suicide attempts, Spijker and colleagues [32] carried out a prospective epidemiologic survey in the 2009 Netherlands Mental Health Survey and Incidence Study. They studied 586 adults with ≥2 depressive symptoms (according to the Composite International Diagnostic Interview-CIDI). The results indicated that the rates of suicidal ideation and suicide attempts in the observed population were, respectively, 16.6% and 3.2%. Several clinical factors seemed to be related to an increased risk of suicide: longer duration (13 months) of depression (*p* = 0.01), anhedonia (*p* = 0.005), feeling useless (*p* = 0.03), anxiety (*p* < 0.01), previous suicidal ideation (*p* < 0.001), and access to professional care (*p* = 0.05). 

Oei and colleagues [34] conducted a cross-sectional study on 46 patients with depressive symptoms recruited from the Department of Psychiatry at the University Hospital in Utrecht (Netherlands). Using several measures of both suicidal ideation and anhedonia, they found that anhedonia scores were positively associated with suicidal ideation scores. Of the anhedonic patients, 71% had suicidal ideation. 

Fawcett and colleagues [16] investigated the possible role of anhedonia in suicidal behavior as a time-related predictor of suicide in major affective disorder, together with other clinical features. In this prospective study of 954 patients affected by major affective disorder, among the nine clinical features studied, the following were correlated with the risk of dying by suicide within 13 months from baseline: anhedonia, panic attacks, severe anxiety, diminished concentration, moderate alcohol abuse, and global insomnia. Long-term suicides (after 13 months from the baseline) were associated with hopelessness and prior suicidal ideation.

### 3.2. Schizophrenic Spectrum Disorders

Jahn and colleagues [35] studied 162 patients affected by schizophrenia or schizoaffective disorder and found that motivation and pleasure scores as rated by the Clinical Assessment Interview for negative symptoms (CAINS) were not directly associated with recent suicidal ideation. However, motivation and pleasure-related scores moderated the relationship between social role functioning and suicide ideation. 

Loas and colleagues [18] investigated the protective nature of negative symptoms by comparing positive and negative symptoms in schizophrenic patients who died by suicide with non-suicidal schizophrenic patients in a prospective study of 150 patients with chronic schizophrenia. They found out that negative symptoms and the deficit syndrome [defined as a subgroup of schizophrenic patients with at least two enduring, primary or idiopathic, negative symptoms (e.g., poverty of speech, restricted affect, diminished emotional range, interests, sense of purpose or social drive) as assessed by the Positive and Negative Syndrome Scale (PANSS) and the subscale of the Brief Psychiatric Rating Scale (BPRS)] were associated with a lower risk of suicide in these patients. The incidence of the deficit syndrome in the suicides (0%) was lower than the incidence in non-suicidal patients (23.5%).

### 3.3. Post-Traumatic Stress Disorder (PTSD)

Spitzer and colleagues [36] conducted a cross-sectional study in a sample of 373 trauma-exposed psychology undergraduate students to examine the relationship between acquired capability for suicide (ACS) and DSM–5 PTSD symptom clusters. Whereas most PTSD symptom clusters had a negative association with ACS, the anhedonia cluster was positively related to both facets of ACS: fearlessness of the pain involved in dying (FOP) and fearlessness about death (FAD). The relationship between anhedonia and ACS may explain the decreased fear of death and the higher levels of pain tolerance that contribute to lethal suicidal behavior in PTSD patients. 

Guina and colleagues [37] studied 480 adults from military medical centers with previous trauma. The PTSD symptoms most strongly associated with suicide attempts were anhedonia, negative beliefs, and recklessness. Suicide attempts were also significantly associated with worse total PTSD symptom severity, childhood physical abuse, the violent death of a loved one, childhood neglect, childhood sexual abuse, substance, alcohol problem, and benzodiazepine prescriptions. 

Chen and colleagues [38] analyzed the data from 36,309 U.S. adults (≥18 years) from the National Epidemiologic Survey on Alcohol and Related Conditions-III and found that both PTSD and anhedonia were associated with attempting suicide.

### 3.4. Other Diagnoses

Hawes et al. [50] explored the different effects of chronic vs. acute anhedonia on suicidal ideation and lifetime suicide attempt history in a prospective study of 395 adult psychiatric outpatient centers in New York City between January 2016 and March 2017. They found out that recent changes in the capacity to experience pleasure was more powerful in predicting near-term suicidal ideation than was chronic anhedonia. 

Winer et al. [11] investigated the association between suicidal ideation and anhedonia in a prospective cohort study of 1529 adult psychiatric inpatients with various diagnoses from DSM-IV using the Beck Depression Inventory II. Anhedonia was associated with suicidal ideation at baseline and at the termination of the study. Changes in anhedonia from baseline to termination predicted a change in suicidal ideation, and anhedonia was a predictor of suicidal ideation regardless of cognitive/affective symptoms of depression. 

Loas and colleagues [40] studied 122 psychiatric patients hospitalized in the psychiatric unit of the CHU of Amiens between 2010 and 2013. They found that suicidal ideation as measured by the Beck Depression Inventory was associated with anticipatory anhedonia as measured by the Physical Anhedonia Scale (PAS) (but not PAS consummatory anhedonia) and with anhedonia as measured by the BDI, but not with scores on the Temporal Experience of Pleasure Scale (TEPS). 

### 3.5. Attempted Suicides

Yaseen and colleagues [39] conducted a cross-sectional study to evaluate the relationship between suicidal behavior and anhedonia, anxiety, feelings of being trapped, and frightened attachment in 135 adults admitted to the Emergency Room of the Mount Sinai Beth Israel Hospital (New York) for a suicide attempt or with high suicidal risk. All participants reported suicidal ideation in the past month. Suicidal behavior was assessed with the Columbia Suicide-Severity Rating Scale (C-SSRS). The results showed that prior suicidal ideation was significantly associated with anhedonia, anxiety, and feelings of entrapment. Entrapment and anhedonia were independently associated also with the severity of suicidal ideation. 

Loas and colleagues [33] examined the correlation between anhedonia and suicide risk in a cross-sectional study of 106 attempted suicides (30 non-depressed and 73 depressed) and 104 normal controls. The participants were evaluated using the PAS and BDI. Depressed attempters were significantly more anhedonic than controls and non-depressed attempters, while no statistically significant difference on the PAS score emerged between healthy subjects and non-depressed attempters. 

Loas [13] followed-up for 6.5 years 106 patients who attempted suicide and who were admitted to the Hospital Nord d’Amiens (between September 1993 and August 1996). During the follow-up period, 6.7% of the patients died by suicide. The patients who completed suicide had a significantly lower score on the PAS as compared to the non-suicide group. Thus, this prospective cohort study indicated that a low level of anhedonia was associated with death from suicide.

### 3.6. Others

In a cross-sectional study, Daghigh et al. [51] tried to replicate findings of the association between anhedonia, suicide ideation ad suicide attempts [19] in an Iranian sample of 404 students at Kashan University (in the center of Iran). In addition to the surprising finding that the Iranian students were more anhedonic than were the American students (more than double), the authors reported that anhedonia was associated with suicidal risk, independently of other symptoms of depression, and also with suicide attempts. 

Loas and colleagues [41] found that anhedonia was related to both suicidal ideation and suicide attempts in a population of 557 physicians. Anhedonia mediated the relationship between suicidal ideation and perceived burdensomeness and thwarted belongingness, variables relevant to Joiner’s Interpersonal Theory of Suicide [44]. Two different features of anhedonia (lack of satisfaction and loss of energy) were associated with suicidal behavior. Lack of satisfaction had an impact on suicidal ideation, while loss of energy had an impact on suicide attempts. 

In a cross-sectional study, Winer and colleagues [19] studied a population of 1122 undergraduate students at a public university in the southern United States and found that anhedonia was associated with suicidal ideation, but not with suicide attempts. 

A cross-sectional study by Loas [42] investigated the association between anhedonia, depression, and suicidal behavior in 224 healthy patients (using the PAS and the BDI) and found that anhedonia scores were not associated with depression scores or current suicidal ideation.

## 4. Discussion

The present review of the literature suggested that findings on anhedonia and suicidal behavior are inconsistent, and it is difficult to propose a theoretical framework that could explain all the research findings. Some studies in this review reported a positive association between the presence of anhedonia and suicidal behavior (high anhedonia scores were associated with suicidal behavior), regardless of the diagnosis of the patients. Other studies found a negative association (low anhedonia scores were associated with suicidal behavior), and a few studies reported no association. The included studies, organized by type of disorder, are summarized in Table 1.

The role of psychiatric diagnosis in the association of anhedonia with suicidal behavior appears to be important. The most consistent association was found for patients with affective disorders (7 of 7 studies reported a significant positive association) and for patients with PTSD (3 of 3 studies reported a positive association). For the two studies of patients with schizophrenia, one found no association (Jahn et al. [35]) and the other found a negative association (Loas et al. [18]). For patients who had attempted suicide (and who were undiagnosed), one study found a positive association [39], one a positive association but only for depressed attempters [33], and one found a negative association [13]. 

The results are also made complex by the type of suicidal risk studied. Winer et al. [19] reported a positive association between anhedonia and suicidal ideation but not for anhedonia and attempted suicide. A meta-analysis by Ducasse and colleagues [45] confirmed that anhedonia scores are higher among patients with than without current suicidal ideation. That meta-analysis evaluated the results of 15 case-control studies, which included 7347 participants (mean age = 30 yrs.) and compared the results for subjects with current suicidal ideation (*n =* 657) vs. subjects without current suicidal ideation (*n =* 6690). Patients affected by different psychiatric disorders (depression; schizophrenia; personality disorders; substance abuse disorder) and subjects without psychiatric diagnosis were included. The results demonstrated that the group with current suicidal ideation presented higher level of anhedonia vs. the group without suicidal ideation (*p* < 0.001; CI = 0.37–0.79). Moreover, the subgroup analysis conducted for a group of patients with homogenous depression scale scores showed that the association between anhedonia and suicidal ideation was not influenced by depression. Those results suggested that people with suicidal ideation had higher level of anhedonia, independently of the presence of depression. For completed suicides, Ballard et al. [31] reported a positive association, whereas Loas et al. [13] reported a negative association. The results may also be made complex by the high incidence of anhedonia in psychiatric patients, as high as 50% as reported in a study from Silverstone [43].

Why is there an association between suicide and anhedonia? The tendency to avoid emotions experienced as unpleasant can be a maladaptive coping strategy [46], and suicidal ideation could also be an avoidance strategy [47,48]. However, avoiding experiences leads to the attenuation of pleasant emotions and the exacerbation of unpleasant ones [49]. The association between current suicidal ideation and anhedonia may also be related to the interaction between the social component of anhedonia (loss of interest in people) and thwarted belongingness, as described by Joiner’s Interpersonal Theory of Suicide [44]. 

This review had several limitations. The studies differed in sample size, study design, and methodology. There were several different measures of anhedonia used in the various studies, and not all of them may be reliable, valid, and equivalent measures. Furthermore, some of the studies did not accurately diagnose the patients, and we have seen in the present review that psychiatric diagnosis may affect the results of the research. Finally, we respected the choice of some authors to use the generic term of “suicidality” [12,32,39,41] to indicate different aspects of the issue, even if we believe that it could be a source of misunderstanding. A more specific term, such as “suicidal risk,” is preferable. 

## 5. Conclusions

The relationship between anhedonia and suicidal behavior is complex. The majority of studies included in our review reported a positive correlation between anhedonia and suicidal risk. The most consistent positive association was found for patients with affective disorders and for patients with PTSD. The positive association was most consistent for suicidal ideation, but less consistent for attempted suicide and completed suicide. However, our review suggests that the assessment of anhedonia may be useful in the evaluation of suicidal risk in depressed patients.

## Figures and Tables

**Table 1 medicina-55-00458-t001:** Summary of reports on suicide behavior and anhedonia.

**Mood Disorder**						
**Reference**	**Study Design**	**Sample**	**Criteria**	**Objectives**	**Methods**	**Results**
Zielinski et al. 2017	Prospective longitudinal study	187 participants	Participants (mean age = 34.41) with a history of non-suicidal self-injury or previous suicide attempts	To verify if modifications in depressive symptoms and anhedonia can be considered as predictive factors of suicide and self-harm behavior	- DASS-21 © to assess depressive symptoms - SLIPS ^¥^ to evaluate anhedonia - SBQ-R ^↓^ to assess several dimensions of suicidal risk - OSI^€^ to evaluate self -harm behavior	Anhedonia scores were higher in those with prior suicide attempts than in those with a history of non-suicidal self-injury. Depressive symptoms predicted suicidal behavior directly and also via mediation through anhedonia
Ballard et al. 2017	Retrospective longitudinal study	100 participants	Patients (18–65 years) affected by treatment-resistant major depressive disorder (*n =* 65) or bipolar disorder without psychotic features (*n =* 35) recruited from several clinical trials of ketamine	To evaluate the relationship between the decrease of suicidal ideation after ketamine administration and the reduction of anhedonia	- BDI *-II and the SHAPS ^#^ to assess anhedonia - SSI5 ^Ø^ to assess suicidal ideation	At baseline, anhedonia scores were associated with suicidal ideation. After administration of ketamine, both suicidal ideation and anhedonia declined. The change in anhedonia accounted for an additional 13% of the variance for changes in suicidal thoughts beyond the effects of depressive symptoms
Ballard et al. 2016	Prospective study from the STEP-BD study (4360 patients)	530 participants	Subjects who attempted or died by suicide (*n =* 103) vs. controls (*n =* 427)	To individualize which clinical dimensions (suicidal ideation, loss of interest, anxiety, psychomotor agitation, high-risk behavior) increase before suicidal conduct	- Clinical Monitoring Form (CMF) for clinic evaluation	Patients with suicidal behavior had an elevation of all symptoms, including loss of interest. Suicidal ideation and loss of interest were highly intense in the months before the suicidal behavior and can be considered a risk factor for suicidal behavior in bipolar patients
Xie et al. 2014	Cross-sectional study	60 participants	Participants (18–60 years) included patients affected by major depressive disorder (*n =* 40) vs. control subjects without psychiatric disorder (*n =* 20). Based on BSS scores, psychiatric patients were divided into two subgroups: high suicidal ideation (HSI) group and low suicidal ideation (LSI) group	To analyze the association between anhedonia, pain avoidance motivation, and suicidal ideation	- BDI *-II to assess depressive symptoms - BSS ^$^ for suicidal ideation - PAS ^%^ and TDPPS ^¢^ to evaluatephysical and mental pain - MID ^~^ and AID ^§^	In AID task, the HSI had longer response times (RTs) under the reward condition than those under the punishment condition (*p* = 0.002); the LSI group and control groups had shorter RTs under the reward condition than those under the neutral condition (*p* < 0.001 and *p* = 0.008, respectively); the LSI group had shorter RTs under the reward condition than under the punishment condition (*p* = 0.003). Pain arousal (*p* < 0.01) and BSS scores were significantly negatively correlated with differences in RTs between neutral and reward conditions and positively correlated with differences in RTs between neutral and punishment conditions. The AID was the best task for detection of hedonic approach to experiences and of pain avoidance so that a reduced motivation to experience anhedonia and an increased one to avoid pain can be considered reliable predictors of suicidal behavior
Spijker et al. 2010	Prospective epidemiologic survey	586 participants	Data were extracted from the Netherlands Mental Health Survey and Incidence Study (NEMESIS). Participants were adults (18–64 years) with a depressive spectrum disorder (≥2 depressive symptoms—according to Composite International Diagnostic Interview—CIDI)	To individuate determinants of suicidality (suicidal ideation and suicide attempts) and compare these two factors	- One item of the CIDI to evaluate suicidal ideation and suicide attempts	Suicidal ideation was influenced by different clinical features: longer duration of depression—13 months—(*p* = 0.01), anhedonia (*p* = 0.05), feeling worthless (*p* = 0.03), comorbid anxiety (*p* < 0.01), previous suicidal ideation (*p* < 0.001), and use of professional care (*p* = 0.05). Anxiety, suicidal ideation, previous suicide attempts, and living alone were significantly related with suicide attempts (respectively *p* = 0.01; *p* = 0.05; *p* < 0.01; *p* = 0.02)
Oei et al. 1990	Cross-sectional study	46 participants	Patients (18–65 years) with depressive symptoms (according to DSM-III) were recruited from Department of Psychiatry of the University Hospital in Utrecht (The Netherlands)	To evaluate the relationship between anhedonia, suicidal behavior, and dexamethasone non suppression	- HDRS ^ß^ and MADRS ^†^ for depression - BSS ^$^ for suicidal ideation - CPAS ° to assess anhedonia - STAI ^℅^ for anxiety -Dexamethasone (DEX) Suppression	The combination of anhedonia (*p* < 0.001), suicidal ideation (*p* < 0.05), and DEX-non-suppression (*p* < 0.05) identified a subgroup of patients affected by depressive disorder who presented three positive symptoms
Fawcett et al. 1990	Prospective study	954 participants	954 subjects (mean age = 38.1) affected by major affective disorder (according to Research Diagnostic Criteria)	To identify clinical factors related to suicidal risk	- Schedules for affective disorders and schizophrenia (SADS)	3% of the subjects committed suicide; 41% of the patients died during the first year of follow-up. Suicide was associated with anhedonia (*p* = 0.005) in the 12-month follow-up
**Schizophrenic Spectrum Disorder**						
**Reference**	**Study Design**	**Sample**	**Criteria**	**Objectives**	**Methods**	**Results**
Jahn et al. 2016	Prospective study	162 participants	Patients (mean age = 46.84) affected by schizophrenia or schizoaffective disorder (according to DSM-IV)	To examine the role of social functioning on suicidal ideation	- BPRS ^↔^ (24-item expanded version) to assess symptoms - RFS ^∑^ and Social Closeness Scale from MPQ - SC ^⌂^ for social role functioning - CAINS ^Ɫ^ to assess negative symptoms	Motivation and pleasure-related negative symptoms influenced the relationship between social role functioning and suicidal ideation (*p* < 0.001). A low level of suicidal ideation was associated with better social role functioning in patients who had low motivation and pleasure-related negative symptoms
Loas et al. 2009	Prospective study	150 participants	150 in- or outpatients affected by chronic schizophrenia were included from April 1991 to July 1995 and followed-up for 14 years	To compare clinical symptoms (positive and negative) of schizophrenic patients who died by suicide vs. patients who died by other causes	- BPRS ^↔^ - CPAS ° to assess anhedonia - BDI *-II - PANSS ^Ꙩ^	Patients who died from suicide had a shorter duration of illness, and a higher level of education vs. patients died from other causes (*p* = 0.02). Negative symptoms and deficit syndrome can be considered as protective factors in schizophrenia.Anhedonia was associated with a higher risk of suicide in schizophrenia (*p* = 0.19)
**Post-Traumatic Stress Disorder**						
**Reference**	**Study Design**	**Sample**	**Criteria**	**Objectives**	**Methods**	**Results**
Spitzer et al. 2018	Cross-sectional study	373 participants	Psychology undergraduates students (mean age = 19.9) enrolled in a public university in the southeastern United States who satisfied criterion A of PTSD (DSM-5)	To analyze the association between PTSD dimensionality (based on the six-factor anhedonia model) and acquired capability for suicide (including fearlessness of pain involved in dying and fearlessness about death)	- FOP ^ж^ and FAD ^∞^ - DSM-5 to assess the diagnosis of PTSD - BDI *-II	Anhedonia cluster symptoms of PTSD was significantly related to the fearlessness of pain involved in dying (*p* = 0.04) and also fearlessness about death (*p* = 0.03)
Guina et al. 2017	Cross-sectional self-report survey	480 participants	Adults outpatients recruited at a military medical center with previous trauma	To investigate the relationships between suicide attempts and PTSD, trauma types, demographics, use of substance and benzodiazepine prescriptions	- Diagnostic and Statistical Manual of Mental Disorders -5(DSM-5) to assess the diagnosis of PTSD and to evaluate suicide attempts, previous traumatic events and abuse of alcohol or drugs	Suicide attempts were significantly associated with severity of PTSD symptom (*p* < 0.0001), childhood physical abuse (*p* = 0.0002), violent death of a loved one (*p* = 0.0037), childhood neglect (*p* = 0.0099), childhood sexual abuse (*p* = 0.0246), substance and alcohol problems (respectively *p* = 0.0189 and *p* = 0.0128), and benzodiazepine prescriptions (*p* = 0.0006)
Chen et al. 2017	Survey	36,309 participants	Participants (≥18 years) were recruited in 2012–2013. A sample of 23,936 subjects (2457 veterans and 21,479 non-veterans) had a history of at least a traumatic event	To evaluate the clinical dimension of PTSD and its relationship with suicide attempts	- Lifetime suicide attempts-AUDADIS -5 ^ꚍ^ to assess the diagnosis of PTSD	PTSD can be considered as a risk factor for suicide attempts and anhedonia. Anhedonia was recognized as a latent factor of PTSD and had a statistically significant effect on suicide attempts (*p* < 0.05)
**Other Diagnoses**						
**Reference**	**Study Design**	**Sample**	**Criteria**	**Objectives**	**Methods**	**Results**
Hawes et al. 2018	Prospective study	395 participants	Participants (≥18 years) recruited from three adult psychiatric outpatient centers in New York City between January 2016 and March 2017	To explore the different effects of chronic vs. acute anhedonia on suicidal ideation and lifetime suicide attempts	- SHAPS ^#^ to assess anhedonia; - BDI *-II to assess depression; - ^℅^ STAI to assess anxiety; - ^$^ BSS and ^ꝸ^ C-SSRS to assess severity of suicidal ideation and history of suicide attempts;	Recent changes in the capacity to experience pleasure is better for predicting near-term suicidal ideation than is chronic anhedonia. The acutely anhedonic group reported greater severity of suicidal ideation in the past-week (*p* = 0.02) and past-month (*p* = 0.04)
Loas et al. 2016	Cross-sectional study	122 participants	122 adult patients (mean age = 44.64) recruited between 2010 and 2013 in the Psychiatric Unit of the CHU of Amiens. Subjects had a diagnosis of mood disorder (*n =* 37) or anxiety disorder (*n =* 85) according to ICD-10. Forty-one patients were admitted for a suicide attempt	To explore the association between anhedonia, alexithymia, impulsivity, suicidal ideation, recent suicide attempt, C-Reactive Protein (CRP) and serum lipid profile	- TAS-20 ꭥ for alexithymia - BIS-10 ךּ for impulsivity - Subscales of the Temporal Experience Pleasure Scale for trait anhedonia - BDI *-II for anhedonia, depression and suicidal ideation - Blood samples for CRP and serum lipid profile (between 7:00 and 8:00 a.m. after at least 12 h of fast)	Anhedonia was related to low total cholesterol level (*p* = 0.043) and low triglycerides level (*p* = 0.077). There was an association between a low level of HDL cholesterol and high suicidal ideation (*p* = 0.015). Patients who committed suicide presented a higher level of CRP vs. patients without suicidal behavior (*p* = 0.033)
Winer et al.2014	Prospective cohort study	1529 participants	Adult inpatients (mean age: 35.55 years) of a private, not-for-profit Psychiatric structure in the southern United States recruited between April 2008 and August 2011. Participants were affected by: depressive disorder (53.0%); bipolar disorder (16.06%); anxiety disorder (4.86%)	To investigate the association between suicidality and anhedonia	- BDI *-II (Anhedonia, depression symptoms and suicidal ideation were assessed at admission and discharge)	Anhedonia was associated with suicidality at baseline (*p* = 0.001) and at the end of hospitalization (*p* = 0.001). Change in anhedonia from baseline to termination predicted a change in suicidality
**Attempted Suicides**						
**Reference**	**Study Design**	**Sample**	**Criteria**	**Objectives**	**Methods**	**Results**
Yaseen et al. 2016	Cross-sectional analysis	135 participants	Attempted suicides or high-risk patients admitted to an emergency room at Mount Sinai Beth Israel Hospital (New York)	To explore the association of anhedonia with suicidal ideation in attempted suicides and high-risk patients	- C-SSRS ^ꝸ^ to assess suicidal behavior - Items 4 and 12 of BDI *-II to assess anhedonia	Prior suicidal ideation was significantly associated with anhedonia, anxiety, and feelings of entrapment. Entrapment and anhedonia were independently associated also with the severity of suicidal ideation
Loas 2007	Follow-up study	106 participants	Attempted suicides admitted to the Hospital Nord d’Amiens	To predict suicide in a sample of attempted suicides	- PAS ^%^ to assess anhedonia	Patients who died from suicide in 6.5-year follow-up had lower anhedonia scores than those who did not die
Loas et al. 2000	Cross-sectional study	207 participants	Depressed patients (*n =* 73) and not-depressed patients (*n =* 30) admitted to the Centre Hospitalier Universitaire (CHU) Nord d’Amiens after a suicide attempt from September 1993 to August 1996 vs. 104 control subjects	To assess the effect of anhedonia on suicide and parasuicide in patients with depressive disorder and in healthy subjects	- CPAS ° to assess anhedonia - BDI *-II to evaluate depression	Level of anhedonia was higher in depressed attempters vs. not-depressed attempters (*p* < 0.001) and vs. controls (*p* < 0.001). There was no difference in PAS score between healthy subjects and non-depressed attempters (*p* = 0.05)
**Others**						
**Reference**	**Study Design**	**Sample**	**Criteria**	**Objectives**	**Methods**	**Results**
Daghigh et al. 2018	Cross-sectional study	404 participants	Students of Kashan University (center of Persia)	To replicate in an Iranian sample finding of the association between anhedonia, suicide ideation and suicide attempts	- SLIPS ^¥^ for anhedonia - CES-D ^ꞔ^ for depression - SBQ-R for suicidal behavior	Anhedonia was associated with suicidal risk (*p* = 0.030), independent of other symptoms of depression, and with suicide attempts (*p* = 0.000)
Loas et al. 2018	Cross-sectional study	557 participants	Participants were physicians	To explore the relationship between suicidality and anhedonia in a general population	- An abridged version of BDI *-13	Anhedonia was associated with both suicidal ideation (*p* = 0.001) and suicide attempts (*p* = 0.001). Anhedonia influenced the relationship between suicidal ideation and perceived burdensomeness (*p* = 0.001) and thwarted belongingness (*p* = 0.001). Suicide attempts were mediated only by thwarted belongingness. “Dissatisfaction” and “work inhibition” (two different component of anhedonia) can influence respectively suicidal ideation (lifetime- *p* = 0.003; recent- *p* = 0.001) and suicide attempts (*p* = 0.001)
Winer et al. 2016	Cross-sectional study	1122 participants	Undergraduate students (18–36 years) of a public university in the Southern United States	To assess the relationship between anhedonia, suicidal ideation, and suicide attempt	- SLIPS ^¥^ for anhedonia - CES-D^ꞔ^ for depression - SBQ-R ^↓^ for suicidal behavior	Suicidal ideation was influenced by anhedonia (*p* < 0.05). Anhedonia was not associated with suicide attempts (*p* = 0.33). Other depressive symptoms presented a correlation with suicide attempts (*p* < 0.001)
Loas et al. 1995	Cross-sectional study	224 participants	Healthy adult subjects of the Picardie (region of France) were included	To investigate the association between anhedonia, depression, and suicidal risk	- CPAS ° to assess anhedonia - BDI *-II to evaluate depression	There was no association between anhedonia and depression or anhedonia and suicidal ideation (*p* = ns)

^§^ AID: Affective Incentive Delay; ^ꚍ^ AUDADIS-5: The Alcohol Use Disorder and Associated Disabilities Interview Schedule; ^*^ BDI: Beck Depression Inventory; ^ךּ^ BIS-10: Barratt impulsivity scale; ^↔^ BPRS: Brief Psychiatric Rating Scale; ^$^ BSS: Beck Scale for Suicide Ideation; ^Ɫ^ CAINS: Clinical Assessment Interview for Negative Symptoms; ^ꞔ^ CES-D: Center of Epidemiological Studies Depression Scale; ^ꝸ^ C-SSRS: Columbia Suicide Severity Rating Scale; ° CPAS: Chapman Physical Anhedonia Scale; ^©^ DASS: Depression Anxiety Stress Scale; ^∞^ FAD: Fearlessness About Death; ^↑^ FDI-24: Future Disposition Inventory-24; ^ж^ FOP: Scales for Fearlessness of Pain Involved in Dying; ^ß^ HDRS: Hamilton Rating Scale for depression; ^†^ MADRS: Montgomery–Asberg Depression Rating Scale; ^~^ MID: Monetary Incentive Delay; ^⌂^ MPQ-SC: Multidimensional Personality Questionnaire; ^€^ OSI: Ottawa Self-Injury Inventory-Clinical; ^Ꙩ^ PANSS: Positive and Negative Syndrome Scale; ^%^ PAS: Psychache Scale; ^∑^ RFS: Role Functioning Scale; ^↓^ SBQ-R: Suicide Behaviors Questionnaire-Revised; ^#^ SHAPS: Snaith–Hamilton Pleasure Scale; ^¥^ SLIPS: Specific Loss of Interest and Pleasure Scale; ^Ø^ SSI5: Scale for Suicide Ideation; ^℅^ STAI: State-Trait-Anxiety Inventory; ^ꭥ^ TAS-20: 20-item Toronto Alexithymia Scale; ^¢^ TDPPS: Three-Dimensional Psychological Pain Scale.

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
