# Peer review of "Can Anhedonia Be Considered a Suicide Risk Factor? A Review of the Literature"

_medicina, 2019, doi:10.3390/medicina55080458_

Round 1
Reviewer 1 Report
interesting review of littérature. Some corrections must be done
1) In the introduction cite one article explaining the discrepancy in the literature on the Relationship between anhedonia and suicide risk (loas, 2014, anhedonia and risk of suicide : an overview, Ritsner (Ed), comprehensive handbook V2, Springer)
2) Fivepoints must be presented that are important to understand the Relationship
- type of anhedonia: state vs trait; recent change of state or not, consummatory vs anticipatory
- type of suicide risk: suicidal ideation, suicide attempt (recent or not), suicide
- type of sample: healthy, psychiatric (what type: depressive...)
- control of confounding variable: age, gender, status (ambulatory, inpatients)
- control of depressive level: effect of anhedonia indepedent of depression
3) Reference lacking: hawes et al, 2018, depression and anxiety, 1-10
daghigh et al, 2018, Suicide and life threatening behavior, April 20.
Author Response
Dear reviewer,
Thank you for your reviews of this MS and for inviting a revision and resubmission. The thoughtful, constructive comments by reviewers and editors were helpful in preparing the attached revision. The following summarizes our responses to each of the comments received. A marked copy of the MS with track changes is also submitted.
1) In the introduction cite one article explaining the discrepancy in the literature on the Relationship between anhedonia and suicide risk (loas, 2014, anhedonia and risk of suicide: an overview, Ritsner (Ed), comprehensive handbook V2, Springer)
Answer: Thanks, we included the suggested article (loas, 2014) in the introduction as useful integration;
2) Fivepoints must be presented that are important to understand the Relationship
- type of anhedonia: state vs trait; recent change of state or not, consummatory vs anticipatory
- type of suicide risk: suicidal ideation, suicide attempt (recent or not), suicide
- type of sample: healthy, psychiatric (what type: depressive...)
- control of confounding variable: age, gender, status (ambulatory, inpatients)
- control of depressive level: effect of anhedonia indepedent of depression
Answer: Thanks for this suggestion, we specified more carefully in the text the relationship between anhedonia and suicide risk considering all the aspects indicated above;
3) Reference lacking: hawes et al, 2018, depression and anxiety, 1-10 daghigh et al, 2018, Suicide and life threatening behavior, April 20.
Answer: Thanks for this integration, we included the suggested articles (Hawes et al, 2018; Daghigh et al, 2018) in the results and in the table, examining in details the main findings in the discussion.
MINOR COMMENTS:
English language and style: Moderate English changes required
Answer: in order to improve the quality of the language throughout the paper, we would like to specify that the manuscript was subjected to a careful editing for language and grammar by a native English.
My coauthors and I hope you will now find the revised MS suitable for timely appearance in Medicina.
Sincerely

Reviewer 2 Report
The authors have written a meaningful qualitative review examining whether anhedonia can be considered a risk factor of suicide. However, there is a recent meta-analysis covering very similar information that is covered in this article. The authors reference that paper (Ducasse et al., 2018) but should more fully unpack what was found and the issues that were reviewed therein.
Also, and most importantly, the authors conclude that the relationship between anhedonia and suicidality "depend(s) on the psychiatric diagnosis of the patient." That does not logically follow from the evidence they have reviewed (nor the available evidence in the literature at large). The authors should either alter this conclusion or explain, in much more detail, why they have made this conclusion.
Author Response
Dear reviewer,
Thank you for your reviews of this MS and for inviting a revision and resubmission. The thoughtful, constructive comments by reviewers and editors were helpful in preparing the attached revision. The following summarizes our responses to each of the comments received. A marked copy of the MS with track changes is also submitted.
1) The authors have written a meaningful qualitative review examining whether anhedonia can be considered a risk factor of suicide. However, there is a recent meta-analysis covering very similar information that is covered in this article. The authors reference that paper (Ducasse et al., 2018) but should more fully unpack what was found and the issues that were reviewed therein.
Answer: even if we have already mentioned this worthy paper (Ducasse et al., 2018), according to your suggestions we decide to emphasize it and to highlight the results in the discussion.
2) Also, and most importantly, the authors conclude that the relationship between anhedonia and suicidality "depend(s) on the psychiatric diagnosis of the patient." That does not logically follow from the evidence they have reviewed (nor the available evidence in the literature at large). The authors should either alter this conclusion or explain, in much more detail, why they have made this conclusion.
Answer: We appreciated your comments about our conclusion and we engaged in clarifying the contradictions reported in the conclusion section of the article;
MINOR COMMENTS:
English language and style: English language and style are fine/minor spell check required
Answer: in order to improve the quality of the language throughout the paper, we would like to specify that the manuscript was subjected to a careful editing for language and grammar by a native English.
My coauthors and I hope you will now find the revised MS suitable for timely appearance in Medicina.
Sincerely
